# ReNeuWell mental well-being app: protocol for a randomised controlled trial

Luke A Egan [1], Justine M Gatt [1,2]

¹Centre for Wellbeing, Resilience and Recovery, Neuroscience Research Australia, Randwick, New South Wales, Australia
²School of Psychology, University of New South Wales, Kensington Campus, Kensington, New South Wales, Australia

**Correspondence to**
Professor Justine M Gatt;
j.gatt@unsw.edu.au

## ABSTRACT

**Introduction** The field of mental well-being interventions includes numerous studies of smartphone app-based programs, but there is a *research-to-retail gap* where many studies pertain to apps that are not publicly available, not used as standalone programs, or not tested in the general population, and many publicly available apps (or their proprietary in-app measures) have yet to be submitted to empirical testing. Furthermore, few well-being apps offer multicomponent interventions, despite such interventions having demonstrated efficacy outside the smartphone context. In response to these openings in the literature and marketplace, we have developed ReNeuWell, an iPhone app designed to measure the user's mental well-being (via the validated Composure, Own-worth, Mastery, Positivity, Achievement and Satisfaction for Well-being (COMPAS-W) scale) and improve their well-being via a personalised, multicomponent program of activities informed by the peer-reviewed evidence base. This article describes the protocol for the preregistered randomised controlled trial (RCT) of ReNeuWell, to test the app in adult participants from the general population of Apple App Store users. It is hypothesised that ReNeuWell users will experience significant increases in mental well-being and decreases in mental distress over the 6–12-week trial period, relative to users of an active control version of the app.

**Methods and analysis** The RCT will recruit participants from Apple Store users who choose to download ReNeuWell in the normal course of browsing the marketplace. Following consent, the app will randomly allocate participants to receive either the full version of the app or the active control version. The full version assesses the user's well-being via the validated COMPAS-W scale, provides feedback on their well-being across six dimensions and creates a personalised schedule of daily positive psychology activities designed to enhance well-being along each dimension. Participants will be instructed to use the app for at least 10 min (at least one activity) per day for the first 6 weeks, and as they wish for the following 6 weeks. Trial outcomes will be measured via in-app surveys administered in weeks 1, 6 and 12. Data collection will begin when the app is officially launched on the Apple Store. Data will be analysed using linear mixed models to estimate condition-by-time interaction effects on the primary and secondary outcomes, and to assess whether any such effects are themselves moderated by other key variables.

## STRENGTHS AND LIMITATIONS OF THIS STUDY

⇒ ReNeuWell is a newly developed, publicly available iPhone app that addresses gaps in both the literature and the market, incorporating a validated well-being measure (the Composure, Own-worth, Mastery, Positivity, Achievement and Satisfaction for Well-being scale) and a personalised, multicomponent program of evidence-based activities and resources.

⇒ The randomised controlled trial of ReNeuWell accounts for the placebo effect by offering an active control condition and includes both a primary endpoint (6 weeks after baseline measures) and a secondary endpoint (12 weeks), allowing follow-up measurements of possible sustained benefits.

⇒ The primary outcomes encompass both mental well-being and psychological distress (depression, anxiety, stress), which reflects the dual-continua conception of mental health.

⇒ The participant screening process excludes those with moderate-to-severe distress, but which may increase the likelihood of ceiling effects (for well-being outcomes) and floor effects (for distress outcomes).

⇒ The study period encompasses a follow-up measure at 12 weeks after baseline, but if efficacy is demonstrated, this period could be expanded to longer timeframes to assess the maintenance of effects.

**Ethics and dissemination** This protocol has been approved by the Human Research Ethics Committee of the University of New South Wales (reference number: HC210302). Trial outcomes will be published in accordance with the preregistered protocol described here, both in the peer-reviewed literature and on the registry website.
**Trial registration number** This protocol was preregistered with the Australian New Zealand Clinical Trials Registry (registration approved on 4 August 2021; trial ID number: ACTRN12621001014842p).

## INTRODUCTION

Optimal mental health is defined not only by minimal symptoms of mental illness but also

by high levels of mental well-being.[1–4] Low well-being, even without accompanying mental illness symptoms, can lead to poor psychosocial outcomes,[1 5] necessitating psychological programs that target well-being specifically, even in non-clinical populations.[6 7] Such interventions are increasingly offered via digital technologies such as websites and smartphone apps,[8–10] some of which have been empirically tested,[11 12] but there remains the *research-to-retail gap*: much of the research pertains to apps that are not publicly available on app stores, have not been tested in the general population of app users and/or are not offered as standalone programs,[13 14] which raises questions of ecological validity. Some studies even pertain to apps that were not given a product name, precluding other researchers from following up on the results. On the market side of the gap, it is common to find long-standing apps whose product descriptions make claims of improved mental health or well-being despite little-to-no empirical evidence,[10 12 15–17] and apps that incorporate proprietary psychological measures (eg, happiness questionnaires) that have not been validated in the peer-reviewed literature. Therefore, despite the large number of mental health apps that have been released over the years,[10 14 15] there are still openings in the market for new, standalone apps that are both evidence-based and publicly available, as well as openings in the literature for peer-reviewed studies of such apps, incorporating established mental health measures.

The research-to-retail gap can be illustrated by considering the studies included in Linardon *et al*'s recent meta-analysis of randomised controlled trials (RCTs) of app-supported mental health interventions.[11] The meta-analysis incorporated 66 effects across 46 studies, finding that app-supported interventions can lead to small-to-moderate reductions in depression, anxiety and stress; and small-to-moderate increases in mental well-being and quality of life. However, few of the 46 papers described interventions that resembled the *typical experience* of downloading a publicly available app from an app store and using it as a *standalone* program without any accompanying resources or interactions.[14] Across the 66 trials, only 14 publicly available mental health apps were examined, and of those apps, only Headspace was evaluated in more than one study. Of the three studies that examined Headspace, only one[18] involved a sample from the general population of adults to which the app is marketed. Only four other apps—Living with Heart,[19] MoodKit, MoodMission and MoodPrism—were studied in samples from the general population, and the latter three were evaluated in the same sample from a single study.[20] Since Linardon *et al*'s[11] meta-analysis, additional RCTs of mental well-being apps have been published (eg, refs.[21–27]), and while it is beyond the scope of this article to provide an exhaustive literature review, we have noted that few of these more recent trials were conducted in the general population of app store users (eg, van Agteren *et al*[26] sampled from the general adult population, although the participants were not recruited 'naturally' through self-initiated downloads

from an app store). Thus, there is actually little research on app-based well-being interventions within ecologically valid contexts. This is not a criticism of any individual study. For example, it is perfectly worthwhile to test whether a given mental health app benefits members of a subpopulation (eg, people with diagnosed major depressive disorder); we are simply noting that such studies are less relevant to our understanding of how such apps affect users in general, to whom all the top mental health apps are marketed.[14]

Although the evidence on app-based well-being interventions is relatively limited, it is nonetheless encouraging. The available research suggests that such interventions can produce significant mental health benefits in terms of both well-being and illness/symptom levels. For example, there is evidence that well-being apps can lead to increases in mental well-being,[19 20 22 23] life satisfaction,[26] positive affect[18 26] and self-compassion[19 28]; improvements in affective balance[23 27]; reductions in depression,[18 20] anxiety,[28] stress[23 27 28] and other psychiatric symptoms[22]; and improved sleep.[22 28] Of course, for a well-being app to have broad impact, it must be not only effective but also used by a sufficiently large number of people. The MIND website (Mobile-health Index & Navigation Database; see refs.[15 29]) provides a public catalogue of mental health apps, informed by the American Psychiatric Association's App Evaluation Model. On 10 February 2025, we searched this catalogue for the term "wellbeing" and related terms such as "wellness", "happiness", "satisfaction" and "positive", retrieving 21 results. After excluding apps that were not designed to increase mental well-being, not designed for the general public and/or not designed to be standalone, self-guided programs, we were left with nine apps whose download numbers were publicly available on the Google Play store (the Apple Store does not display such data). All nine apps had been downloaded at least 100 000 times. Although downloading an app does not imply that the user will actually use the app extensively,[30] these figures illustrate the demand for app-based well-being interventions and the feasibility of their delivery at scale.

For a new well-being app to be potentially beneficial, it should offer something beyond the contents of existing apps, while being grounded in an established model of mental well-being. A recent narrative review[14] examined the top 20 mental health apps on the Apple and Google stores, which together accounted for 98% of monthly active users (MAUs, ie, unique individuals who used a mental health app at least once in the 1-month reference period). Of these apps, 18 provided unguided, standalone programs, most of which fell into one of three categories: mindfulness/meditation, journaling/self-monitoring or conversing with an artificial intelligence chatbot. The top two mindfulness/meditation apps—Headspace and Calm—together accounted for approximately 70% of MAUs. Clearly, those seeking mental health support via smartphone apps either prefer mindfulness/meditation exercises or have few alternatives. The latter is plausible: Lagan *et al*'s recent review of 278 mental health apps[15]

found that the three most common activities (provided by hundreds of different apps) were self-monitoring, journaling and mindfulness exercises. In contrast, exercises from evidence-based therapies, such as cognitive behavioural therapy (CBT) or acceptance and commitment therapy (ACT), were offered by far fewer apps (<60 for CBT and <20 for ACT). Thus, there may be unmet demand for new apps that provide activities and resources not included in many (if any) of the current offerings. The positive psychology literature describes numerous exercises and strategies to raise mental well-being whose efficacy has been demonstrated in non-app interventions[6 7 31]; these are prime candidates for conversion to app-based formats where possible. Lagan *et al*'s[15] review also highlighted "the lack of comprehensive apps that would facilitate multiple diverse uses" (p. 8). Although a user could download multiple apps to cover a range of activities, there are numerous drawbacks to this approach,[17] including the obvious inconvenience, the possibility of receiving contradictory notifications or advice from two or more apps, and (with paid apps) the expense.

Furthermore, van Agteren *et al*'s recent meta-analysis of mental well-being interventions[7] concluded that the two kinds of intervention most consistently associated with positive outcomes were mindfulness-based activities and *multicomponent* positive psychology interventions (comprising two or more positive psychology techniques in a single program). Although this meta-analysis was not limited to app-based or digital interventions, it clearly suggests that multiple activities should be considered in programs to improve mental well-being. Despite this, a recent review of 19 popular mental health apps[32] found that 8 of the apps offered only a single evidence-based treatment component (eg, meditation only), with an average of 2.32 components per app across all 19 apps, none of which offered more than 5 components (for comparison, van Agteren *et al*'s meta-analysis covered over 15 types of positive psychology intervention). Moreover, Marshall *et al*[33] have observed that many mental health apps "only offer singular or novelty interventions that do not qualify as comprehensive therapeutic treatments and/or diagnostic instruments" (p. 217).

Given the preceding considerations, we have developed the forthcoming iPhone app 'ReNeuWell' (a portmanteau of 'Resilience, Neuroscience, Wellbeing'), which may address the opening for a standalone, publicly available, multicomponent well-being app for the general adult population. ReNeuWell comprises psychoeducational materials, positive psychology exercises and numerous other activities and resources, with the aim of increasing mental well-being in users. We included as many evidence-based activities and resources as possible, within the inherent limitations of app-based delivery as well as budgetary and logistical constraints. To track each user's well-being and provide them with a personalised program, ReNeuWell incorporates a comprehensive measure of mental well-being—the Composure,

Own-worth, Mastery, Positivity, Achievement and Satisfaction for Well-being (COMPAS-W) scale[5]—which has been validated in numerous prior studies with regard to both its psychometric properties and various correlates, including health behaviours, work productivity, genetics, neural activity and cognitive functioning.[3 5 34–39] Indeed, the entire app is grounded in the COMPAS-W model of well-being. Each activity is designed to target one of the six dimensions of well-being measured by the COMPAS-W, and the psychoeducational materials educate the user on the six dimensions and how each contributes to total well-being. In short, ReNeuWell addresses key limitations of many previous app-based mental health interventions. It offers a multicomponent program, tracks user outcomes via a validated well-being measure, its activities are explicitly based on the peer-reviewed literature and allows our proposed RCT to have greater ecological validity because the app provides a standalone, self-guided intervention for smartphone users in the general adult population, who will access it via self-initiated downloads from the app store.

Given our target population, it is worth noting both that there is a need for greater mental well-being in the general adult population and that this outcome can indeed be achieved via the kinds of positive psychological techniques offered by our app. As mentioned above, mental health is defined by both illness symptoms and levels of well-being. These domains comprise *dual continua*, accounting for approximately 28% of shared variance[2] Thus, the outcomes are largely independent, so treating mental illness does not imply that mental well-being will necessarily improve, and vice versa. A substantial body of research has shown that even among those without symptoms of mental illness, levels of mental well-being usually fall into the 'languishing' or 'moderate' categories, with only a minority in the 'flourishing' category[1–4 40]; thus, there is considerable scope for improvement. This is important not only because well-being is valuable in itself, but also because well-being predicts a range of other positive outcomes. Numerous studies have indicated that mental well-being promotes workplace productivity,[41 42] cognitive performance,[43 44] social functioning[42–44] and physical health and longevity.[42–47] Furthermore, high levels of current well-being can predict a lower incidence of depression up to 10 years later, even after controlling for prior depression levels,[48] which illustrates the modest although significant shared variance between well-being and mental illness. Positive emotion and subjective well-being can also promote mental resilience to stress and trauma.[42] Fortunately, RCTs of mental well-being programs[7] and positive psychology interventions[49] have shown benefits in both clinical populations and healthy participants from the general population, with outcomes including not only higher mental well-being but also greater quality of life, higher levels of character strengths and lower levels of depression, anxiety and stress.[49] These studies involved many of the positive techniques and strategies that we have included in our newly developed well-being app.

ReNeuWell was invented by the present authors alone, with the software programming and graphic design implemented by the digital development studio Miroma Project Factory. Here, we describe the protocol for a preregistered RCT of ReNeuWell. We hypothesise that the app will produce significant improvements in mental well-being and reductions in psychological distress, relative to an active control version of the app.

## METHOD

### Study design

This protocol is for a single-blinded, active-controlled RCT with two conditions (training vs active control) and three measurement occasions (surveys in weeks 1, 6 and 12). The 6-week and 12-week endpoints were selected because we have previously found significant effects from a smaller set of similar activities after 6 weeks.[31]

### Methods and procedure

#### The ReNeuWell app

ReNeuWell will be commercially available on the Apple Store to the general public. After downloading the app and opening it for the first time, users will receive the option to continue as an ordinary paying user or to enrol in the RCT and receive free access for the 12-week study period. Thus, everyone who downloads ReNeuWell will be invited to participate. Those who select the paid version will proceed to the full version of the app and will not be included as participants. Those who select the RCT option will proceed to the Participant Information Statement and Consent Form (see online supplemental material), followed by screening questions to confirm eligibility. If they provide consent and pass the screening questions, participants will be randomised automatically by the app (via alternating allocation with a 1:1 ratio) to either the training condition (providing the full version of the app) or the active control condition (providing a dummy version of the app). Following randomisation, users will create an account and proceed to the app proper.

ReNeuWell comprises two main components: a recurring survey and a suite of activities. The survey constitutes the COMPAS-W scale,[5] which measures the user's well-being along six dimensions: Composure (dealing effectively with stress or adversity), Own-worth (a sense of autonomy and self-respect), Mastery (self-confidence, perceived control over one's environment), Positivity (happiness, optimism), Achievement (setting and pursuing goals) and Satisfaction (with one's life, health and relationships). On first using the app, and every 6 weeks thereafter, participants complete the COMPAS-W, and their scores on the six dimensions are used by the app to generate a personalised program of daily activities for the next 6 weeks. Users are not shown their exact numeric scores on the COMPAS-W, but rather a screen titled *Your Results*, which provides both visual and written feedback. The written feedback comes in one of three

**Table 1** List of activities offered in the full version of ReNeuWell, along with citations of peer-reviewed studies that informed their development or support their inclusion in the app

| Title/topic | Citations |
|---|---|
| Acts of kindness | 31 64–67 |
| Assertiveness training | 68–72 |
| Body language | 73–76 |
| Building strengths | 6 77–79 |
| Colouring mandalas | 80–83 |
| Controlled breathing | 84–88 |
| Facial expressions | 89–92 |
| Gratitude diary | 6 77 93–97 |
| Guided meditation | 86 98–103 |
| Mindfulness | 98 101 104–110 |
| Personal values | 111–114 |
| Positive event scheduling | 115–121 |
| Positive reminiscence | 31 122–126 |
| Recent achievements | 127–131 |
| Self-compassion | 31 132–137 |
| Setting goals | 138–144 |
| Showing empathy | 65 145–149 |
| Stress and coping | 150–154 |

forms, corresponding to the three categories of mental health described by Keyes[1] [2] and established for the COMPAS-W[5]: *languishing, moderate well-being* and *flourishing*. Participants in the RCT will also complete additional measures within the recurring surveys. However, those in the active control condition will not be shown the *Your Results* screen.

After viewing their COMPAS-W results, participants in the training condition will be informed that the app will now generate their personalised program of activities, which constitutes the second main component of ReNeuWell. The app currently offers 18 activities, some of which come in multiple versions to cover different aspects of the relevant topic. Depending on the study outcomes, future updates to the app may add new activities or variations of existing ones. Table 1 lists each activity, along with citations of peer-reviewed studies that informed its development or support its inclusion in the app. In contrast, the active control participants will not be told that they will receive a personalised program because the reduced app contains only a single program with three activities, designed to be neutral with regard to mental well-being (ie, harmless but not beneficial). These three activities are named Acts of Novelty, Life Reminiscence and Self Esteem, and are based on the active control conditions implemented and tested in an RCT of an online well-being intervention.[31]

Once a participant has received their program, every time they open the app (for the next 6 weeks), they will

be taken to the *dashboard* (the 'hub' screen of the app), from which they can browse their schedule of activities and select one or more to complete each day. By default, the app generates a reminder notification for users who have not opened the app for more than 72 hours, reminding them to use the app and suggesting that they try another activity. ReNeuWell also promotes engagement by allowing the user to link the app with their iPhone calendar, to record events or reminders related to their well-being activities. For example, for the 'positive event scheduling' activity, the user can add positive events to their calendar directly from the app.

The dashboard also provides access to additional resources and features, including psychoeducational materials on mental well-being (with the active control app containing fewer psychoeducational resources). All users can also access contact information for mental health support services (eg, 24-hour crisis hotlines), report an issue (eg, a software glitch) or withdraw from the RCT via the dashboard. Participants can withdraw at any time without penalty and without having to provide a reason. Those who withdraw will have the option of continuing as a paying user. The contact information for mental health support services is also provided in the Participant Information Statement provided to all prospective participants during the informed consent process.

### Instructions and timeline

Participants in both conditions will be instructed to use ReNeuWell for at least 10 min per day for the first 6 weeks of the 12-week study period. After week 6, participants are instructed to use the app as they see fit and informed that they will receive their final survey in week 12. For this reason, week 6 is the primary endpoint of the study, with week 12 as the follow-up. The study period is similar to those of previous RCTs of mental health apps, which have shown benefits within durations ranging from 7 days to 16 weeks.[18–20 22 23 26 27] Furthermore, it is especially important for app-based interventions to produce effects within relatively short timeframes because if the user does experience subjective benefits soon after downloading an app, they may delete it and try one of the many alternatives.

In both conditions, each participant's schedule of daily activities is organised into weeks, with participants able to see the current week of their 6-week program and any previous weeks if applicable. Participants can navigate to earlier weeks if they wish to complete an activity that they skipped, but they cannot access future activities ahead of schedule. In order for the next week of the program to be unlocked, the user must open the app at least once during the current week. Therefore, although our RCT is designed as a 12-week study, it will be possible for participants to take longer than 12 weeks to complete their programs. However, the app automatically records when a given activity or survey has been completed, allowing us to control statistically for any divergences from the intended 12-week schedule. After completing the week 12 survey and activities, training participants will be offered

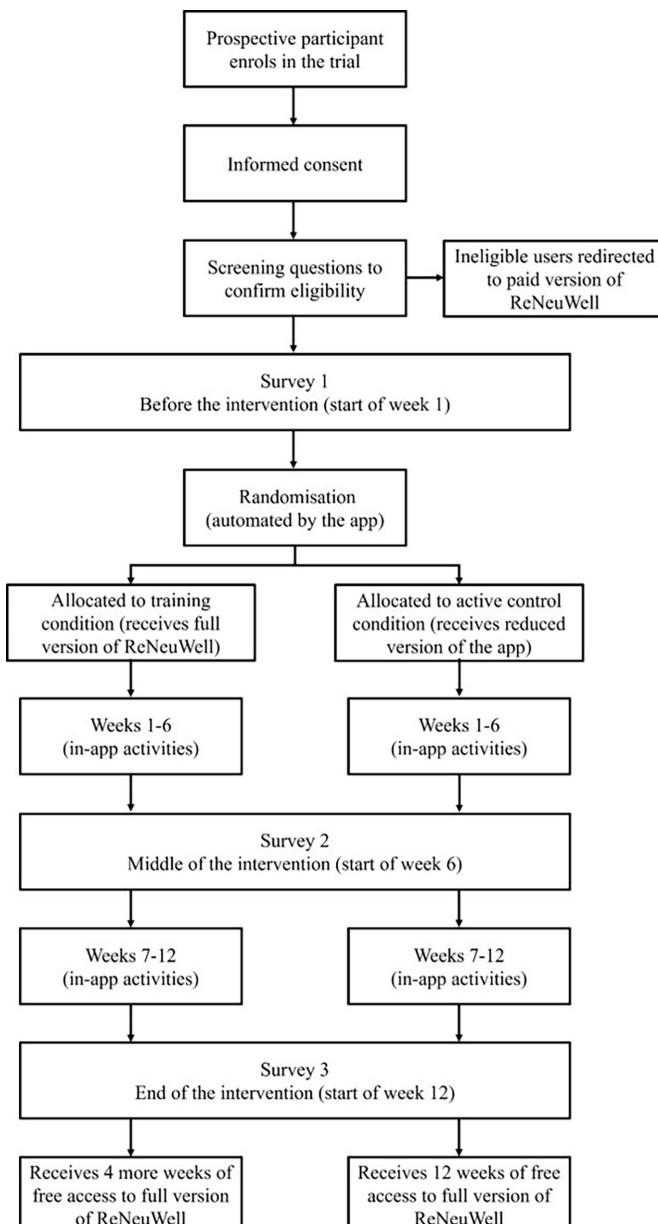

**Figure 1** Study design flow chart

an additional 4 weeks of free access to the app (the full version this time) while control participants will be offered 12 weeks of free access to the paid version. Thus, control participants will not know that they received a reduced version of the app until they unlock the full version after week 12. Indeed, the Participant Information Statement informs prospective participants that they will be randomly allocated to one of two versions of the app, but they are not told that one of the versions is a reduced version and will not be told which version they received. Each participant's app simply appears to them under the name 'ReNeuWell'. The study timeline is illustrated in figure 1.

## Participants

### Recruitment

The participants will be recruited from the general population of iPhone users who choose to download ReNeuWell from the Apple Store. The RCT will not happen within a fixed calendar period; rather, prospective participants will be invited to participate in an ongoing manner, whenever they first open the app. All aspects of each condition (such as instructions, activities and educational materials) will be delivered purely via the app. Therefore, participants will not need to interact with another app, a website or any other entity (including the researchers) at any stage of the study. On the ANZCTR registry website, the anticipated date of first participant enrolment is listed as 1 February 2025. Recruitment will continue until the target sample size has been reached.

### Sample size

The minimum sample size was calculated with the software package G*Power (V.3.1.9.2).[50 51] The type 1 error rate was set at the conventional level of 0.05. With the desired power of 95% and assuming a correlation of 0.5 between the repeated outcome measures, our calculations determined that a total sample size of 324 participants would be needed to detect a small group-by-time interaction effect (partial $\eta^2$=0.01) with two conditions and two measurement occasions (our study has three measurement occasions, but the second occasion is the primary endpoint of the study, and a power calculation with two occasions is stricter than one with three). However, given that app-based interventions demand relatively low commitment from participants, it is reasonable to assume an attrition rate of at least 50% by the second measurement occasion. Indeed, a recent meta-analysis of RCTs of smartphone interventions[52] found attrition rates of 53.4% between weeks 9 and 12. Therefore, we set our target sample size at 486 participants, with 243 in each condition.

### Inclusion criteria

Prospective participants will be informed that participation in the RCT is open to those who are willing to use the app for at least 10 min per day for at least the first 6 weeks of the 12-week study period (and up to 12 weeks), and willing to complete a 10 min survey in weeks 1, 6 and 12. Participants will also need to meet these additional eligibility criteria:

Aged 18 years or older (for informed consent). Able to understand written English (ReNeuWell is currently available in English only), Absence of moderate-to-severe symptoms of mental illness, as measured by a single-item question and scores on the Kessler Psychological Distress Scale—10-item version (K10) screening questionnaire.[53] Residing in one of the following countries: Australia, New Zealand, the UK, Ireland, the USA, Canada (English-speaking countries). Possesses an iPhone (or iPod Touch) with operating system iOS 14.0 or later (although we intend to release the app on Android devices in the future, pending funding).

When ReNeuWell is first released on the Apple Store, participation in the RCT will be an option only for users residing in Australia. We have approval to recruit participants in all six of the aforementioned countries, but we have decided to confine the study to Australia initially, so that we will be well-positioned (being within our home country) to address any unexpected issues with the app or the trial. Barring any major technical issues or adverse events in this initial period, participation will be extended to the other five countries (and further countries pending future developments).

### Screening questions

The screening questions ask for the user's age, whether they understand written English, which country they reside in and whether (yes/no) they currently suffer from a psychological or psychiatric condition. In addition, the screening questions include the K10.[53] K10 scores of 25 or higher are considered indicative of moderate-to-severe distress and therefore of the respondent being ineligible for study participation. Those who do not pass the screening questions will be redirected to a screen whereby they can continue using the app as a paying user, if they wish. The screening questions (except the K10) are provided in the supplementary materials.

## Survey measures

At each measurement occasion (ie, when the participant first opens the app in weeks 1, 6 and 12), the participants will complete a 10 min survey and will not be able to access further activities until the survey is complete. Each survey comprises a range of questionnaires, but not all measures will be administered at each measurement occasion. Table 2 outlines the questionnaire timings.

### K10

K10[53] will be administered during the initial screening process (as already described) and also at the beginning of the second and third surveys. A moderate-to-high K10 score at the screening stage will make the user ineligible for further participation, with K10 scores again assessed at the second and third survey timepoints. The K10 comprises 10 questions on various symptoms of psychological distress (eg, depression and anxiety symptoms), with a 5-point response scale ranging from *none of the time* to *all of the time*. Respondents are instructed to answer the questions in terms of how they have been feeling *over the past 30 days*. K10 has exhibited acceptable reliability and validity in previous studies, and is one of the most frequently used measures in mental health research and practice.[54–56]

### Demographics, Health and Lifestyle questionnaires

The Demographics questionnaire will ask about characteristics, including marital status, highest education and current occupation. The Health questionnaire will ask about the user's psychiatric history, including whether they have been diagnosed with and treated for a mental illness in the past. The Lifestyle questionnaire will ask

Table 2  Questionnaires delivered at each timepoint

|  | Screening (start of week 1) | Survey 1 (start of week 1) | Survey 2 (start of week 6) | Survey 3 (start of week 12) |
|---|---|---|---|---|
| Screening questions | ✓ |  |  |  |
| K10 | ✓ |  | ✓ | ✓ |
| Demographics |  | ✓ |  |  |
| Health |  | ✓ |  |  |
| Lifestyle |  | ✓ |  |  |
| COMPAS-W |  | ✓ | ✓ | ✓ |
| DASS-21 |  | ✓ | ✓ | ✓ |
| HPQ |  | ✓ | ✓ | ✓ |
| App satisfaction |  |  | ✓ |  |
| Adverse event assessment |  | ✓ | ✓ | ✓ |

COMPAS-W, Composure, Own-worth, Mastery, Positivity, Achievement and Satisfaction for Well-being ; DASS-21, Depression, Anxiety and Stress Scales—21-item version; HPQ, Health and Work Performance Questionnaire; K10, Kessler Psychological Distress Scale—10-item version.

about the user's consumption of caffeine, cigarettes and alcohol, as well as their diet, exercise and sleep habits. The Demographics, Health and Lifestyle questionnaires are provided in the supplementary materials.

## COMPAS-W scale

The COMPAS-W Scale[5] is a 26-item measure of overall mental well-being and the six subcomponents of Composure, Own-worth, Mastery, Positivity, Achievement and Satisfaction. Each item is a statement with a 5-point response scale ranging from *strongly disagree* to *strongly agree*. The respondents are instructed to answer each statement in terms of how they feel *most of the time* for the week 1 survey, and then how they felt *over the past month* for later timepoints. The COMPAS-W scale has exhibited acceptable reliability and validity in several previous studies.[5 34–37 57]

## Depression, Anxiety and Stress Scales—21-item Version

The Depression, Anxiety and Stress Scales—21-item version (DASS-21)[58 59] is a measure of psychological distress (ie, symptoms of depression, anxiety and stress) comprising 21 statements with a 4-point response scale ranging from *did not apply to me at all* to *applied to me very much or most of the time*. The respondents are instructed to answer each statement in terms of how much it has applied to them *over the past week*. The DASS-21 has exhibited acceptable reliability and validity in prior non-clinical research.[60 61]

## Health and Work Performance Questionnaire scales (employee version)

Two sections from the employee version of the WHO Health and Work Performance Questionnaire[62 63]—scale B9 and scale B12—will be used to measure participants' work performance. Scale B9 comprises five items asking how many days in the past 4 weeks the respondent missed an entire workday or part of a workday (for health or other reasons), or performed work outside their normal working hours. Scale B12 contains seven items assessing the respondent's appraisal of their own work performance in the past 4 weeks, with a 5-point response scale ranging from *none of the time* to *all of the time*.

## App Satisfaction Survey

In the week 6 survey, participants will receive the ReNeu-Well App Satisfaction Survey, containing a range of questions regarding the participant's opinion of the app and what they liked or disliked about it. This questionnaire is provided in the supplementary materials.

## Adverse Event Assessment

Each survey will conclude with an Adverse Event Assessment, constituting a text box in which the participant can report any aspect of the app or survey that caused discomfort or distress. If the participant has no adverse events to report, they will leave the box empty and finish the survey. This section of the survey also provides the following reminder: "In the App Settings menu, we have provided contact details for Mental Health Support Services for your country, should you ever feel distressed and require additional mental health support." The Adverse Event Assessment is provided in the supplementary Materials.

## App usage metrics

Data regarding participant usage metrics (eg, which activities they completed, how much time they spent using the app) will be recorded within the app and via Google back-end software (Firebase).

## Anonymity and confidentiality

All data collected by ReNeuWell will be automatically uploaded to a secure, dedicated server (located in Australia). On this server, users' account data (ie, names

and email addresses) are stored separately from their survey data and are only accessible using a permissions-based system. This ensures security of the data over and above encryption. The survey data will not include any personally identifying information but rather a trial participant ID number that is not shared with the user. Only the research team and app development team (Miroma Project Factory) will have access to the server. The de-identified data will then be downloaded directly from this server onto the secure internal server at Neuroscience Research Australia (NeuRA) for storage and analysis. Only the research team will have access to the data.

## Data availability

Per ethical requirements and participant consent, data will not be shared beyond the research team.

## Statistical analysis

The primary outcomes for this RCT will be the participants' levels of both mental well-being (as measured by the COMPAS-W) and psychological distress (as measured by the DASS-21). The secondary outcomes will be psychological distress as measured by the K10 and work performance. Potential moderator variables include levels of usage and enjoyment of the app, past mental illness, lifestyle habits, baseline mental health levels and demographic characteristics such as age, sex, education, or occupation.

The initial analyses will be conducted on an intention-to-treat basis and followed by per-protocol analyses. Missing values will be estimated via multiple imputation. The hypotheses will be tested via a linear mixed model estimating the group-by-time interaction effects on the outcome variables. It is hypothesised that levels of mental well-being (COMPAS-W) will increase and levels of psychological distress (DASS-21) will decrease in the training condition relative to the control condition. We will also test for group-by-time interaction effects on work performance and the K10 measure of psychological distress. Any interaction effect may itself be moderated by variables such as age, sex, education, occupation, psychiatric history, lifestyle habits, app usage metrics or baseline mental health levels. Furthermore, we will assess user engagement and retention by computing statistics on the number of participants, the levels of app usage and the levels of attrition relative to the overall number of app downloads. We will also use week 1 and 6 survey responses to examine possible differences between participants who withdraw after these timepoints and those who remain in the study until the week 12 survey, allowing us to interpret our findings with respect to any such differences.

## Ethics and dissemination

This protocol has been approved by the Human Research Ethics Committee of the University of New South Wales (reference number: HC210302) and preregistered with the Australian New Zealand Clinical Trials Registry (registration approved on 4 August 2021; trial ID number: ACTRN12621001014842p). Our findings will be published in accordance with this protocol, both in the peer-reviewed literature and on the registry website.

## Public involvement

Because ReNeuWell is designed for use by the general adult population, it was appropriate to conduct several rounds of consultations with members of the general public throughout the app's development. During initial design stages, we conducted user acceptance testing with volunteers from the general public, who provided feedback on the app's design, colours, contents and features. The app was subsequently refined over numerous iterations, after which beta testing was conducted with local community members for feedback on usability and acceptability of the app, including identification of any bugs or feature issues. The app was then refined further and retested in-house across multiple iterations, before being 'soft launched' on the Apple Store for additional testing by a new group of local community members (ie, volunteers could download ReNeuWell from the Apple Store for testing purposes).

## DISCUSSION

The field of app-based interventions to improve mental well-being has yielded promising findings to date, but there remains a *research-to-retail gap*,[13 14] where many studies pertain to apps that are not publicly available, have not been tested in the general population and/or contain proprietary well-being measures that have not been validated in the peer-reviewed literature. Furthermore, few previous studies have been conducted in an ecologically valid setting, where a user freely chooses to download a publicly available app from a mainstream app store and then uses the app as a standalone program without ever needing to interact with another person or entity. On the market side of the gap, many popular apps have remained available on mainstream app stores for years without ever being submitted to peer-reviewed empirical assessment, and some of these apps have product descriptions that make claims of efficacy despite the lack of empirical evidence.[10 12 15 33]

Alongside the research-to-retail gap, another limitation is that most well-being apps cover only one or a few techniques.[32 33] The literature suggests that mindfulness exercises, which are among the techniques most commonly offered by well-being apps,[14 15] can indeed be effective for increasing well-being,[7] but multicomponent interventions (comprising a range of positive psychology exercises) also have efficacy[7] but have not been offered by many apps previously.[32] Furthermore, although some well-being apps offer evidence-based exercises,[32] few if any are informed by a comprehensive, empirically validated model of well-being. The COMPAS-W framework[5] provides not only a multidimensional description of total well-being, but also a validated scale to measure each dimension and overall well-being.

ReNeuWell is a newly developed iPhone app, grounded in the COMPAS-W model of mental well-being,[5] which addresses these openings in the literature and in the marketplace, and its evaluation in our upcoming RCT will hopefully address some of the key gaps in our knowledge of mental well-being apps and their effectiveness. ReNeuWell provides a tailored program of daily activities and other resources designed to increase well-being in one or more of the six COMPAS-W dimensions. We aim to determine whether this publicly available, standalone app can promote greater well-being and reduce distress in the general adult population of Apple Store customers who choose to use it.

**Acknowledgements** The app was developed by Miroma Project Factory in consultation with the authors.

**Contributors** LAE conducted the literature review, drafted the initial article and subsequent versions, and contributed to the original copy of the app. JMG obtained the grant funding to support the app build, conceptualised the original ReNeuWell app and its design (including copy content), formulated the clinical trial study design and aims of the article, and helped edit the original draft and revised versions. Both authors revised and edited the final version of the article.

**Funding** Costs to build the app and run this project were funded by a Mindgardens Neuroscience Grant awarded to JMG. LAE was supported by a Mindgardens Neuroscience Grant awarded to JMG, and a National Health and Medical Research Council (NHMRC) Project Grant (1122816) awarded to JMG. JMG was supported by a National Health and Medical Research Council (NHMRC) Project Grant (1122816). The study funders had no role in designing the present protocol, nor will they have any role in data collection, management, analysis, interpretation or dissemination.

**Competing interests** LAE, JMG, NeuRA and UNSW NSi will receive proportional payments from any net income received from paying users of the ReNeuWell app. The trial has been preregistered with ANZCTR, and all trial outcomes will be reported in accordance with the published protocol. All trial outcomes will be made publicly available, including null results, which will be used to improve the app for future use and potential retrials, pending funding availability.

**Patient and public involvement** Patients and/or the public were involved in the design, or conduct, or reporting, or dissemination plans of this research. Refer to the Methods section for further details.

**Patient consent for publication** Not applicable.

**Provenance and peer review** Not commissioned; externally peer reviewed.

**ORCID iDs**
Luke A Egan http://orcid.org/0000-0003-0544-8851
Justine M Gatt http://orcid.org/0000-0002-9276-6358

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
