## [Reviewer comments · BMJ Open]

ARTICLE DETAILS

Title (Provisional)

The ReNeuWell mental wellbeing app: Protocol for a randomised controlled trial

Authors

Egan, Luke Anthony; Gatt, Justine M.

VERSION 1 - REVIEW

Reviewer	1
Name	Strudwick, Gillian
Affiliation	Centre for Addiction and Mental Health, Information Management Group
Date	08-Dec-2024
COI	None

Thank you for providing the opportunity to review this interesting protocol on the ReNeuWell app. I have several points that I think the authors may find useful to discuss in their protocol.

- 1) Given that most apps never get used, how will the investigators ensure its used for this study? How will this be managed, and engagement encouraged? I am not convinced thus far that this study is feasible given this big challenge in the app space.
- 2) Given the first point re: app non-usage, I think a stronger argument needs to be made about why creating an app in the first place is a valuable endeavor. Is there prior work that suggests that uptake of something like this could occur at scale and be valuable. This could be further developed in the introduction.
- 3) The authors could clarify who the target population is, and if there is a need for 'mental wellness' among this group. While there is certainly a great need for mental health services, this app is not aimed at doing this. Without this, the paper isnt providing a convincing argument that this is a worthwhile endeavor.
- 4) For those who download and decide to use the app, what forms of engagement strategies might there be to ensure continuous use e.g. push notifications?

5) The authors could also describe what do they do if people answer with low scores on depression, anxiety and stress scales, and how they manage risk here.

Very interesting evaluation approaches for apps are proposed.

I hope the authors find this useful.

Reviewer	2
Name	Kim, Myoungsuk
Affiliation	Kangwon National University, College of Nursing
Date	07-Jan-2025
COI	None

It was a great pleasure to review your manuscript, which presents valuable insights and contributes significantly to the field. I have provided detailed comments and suggestions throughout the manuscript, which I hope will help further enhance the quality, clarity, and impact of your research.

1. Abstract: It would be beneficial to include a description of the ReNeuWell App in the abstract.
2. Introduction: The review of prior research on mental wellbeing apps is well-conducted. However, while the limitations of existing apps are described in relation to the development of ReNeuWell, is there any explicit mention of how the newly developed app addresses these limitations? Although the app is differentiated from existing ones, it doesn't seem to clearly overcome the shortcomings of previous apps. Please elaborate on the unique features of this app and how it differentiates itself from other apps.

3. Methods

1. The anticipated attrition rate is over 50%, which could impact the reliability of the study's results. It would be important to explore strategies to reduce the attrition rate. Please include potential measures to address this issue.
2. The study will last for 12 weeks, but it might be necessary to provide justification for this duration. 12 weeks may not be sufficient to fully assess improvements in mental health.
3. Since only participants without mental health issues are included in this study, there may be limitations in evaluating the actual effectiveness of the app. Specifically, because participants do not have mental health problems, the effects of the app may be underestimated or deemed ineffective. It may be beneficial to include participants with mental health conditions to better assess the app's effectiveness.

VERSION 1 - AUTHOR RESPONSE

Reviewer 1:

Dr. Gillian Strudwick, Centre for Addiction and Mental Health

Comments to the Author:

Thank you for providing the opportunity to review this interesting protocol on the ReNeuWell app. I have several points that I think the authors may find useful to discuss in their protocol.

1) Given that most apps never get used, how will the investigators ensure it's used for this study? How will this be managed, and engagement encouraged? I am not convinced thus far that this study is feasible given this big challenge in the app space.

We are not aware of any research to suggest that most apps never get used. We tried to find relevant information online, but credible sources were lacking (as far as we could tell). We could not find relevant statistics through ordinary Google searches, but we noted that various AI tools (e.g., Google's AI overview, Microsoft's Copilot) claimed that most apps are never used, but when we followed up on their cited sources, we found that the sources did not provide relevant statistics to support this claim. Interestingly, OpenAI's ChatGPT does not endorse the claim, stating instead that specific data are unavailable. Taking the Google Play store as an example (because download numbers for the Apple App Store are not publicly available), ChatGPT claimed that the store contains approximately 2 million apps, of which a minority (approximately 200,000) have been downloaded at least 10,000 times. However, it was not able to state how many apps have been downloaded at least 500 times (our target sample size is 486). Interestingly, Google's AI overview contradicted itself on this issue, stating that "estimates suggest that a substantial majority of apps on the Play Store would fall into this category" (i.e., at least 500 downloads), but again, the cited sources lacked relevant statistics. Our Google searches returned statistics on a related issue: the proportion of apps that are used only once after being downloaded. These statistics (from <https://www.statista.com/statistics/271628/percentage-of-apps-used-once-in-the-us/>) suggest that approximately 25% of apps are used only once after being downloaded. It is also worth noting that statistics on app usage may be distorted by so-called "shovelware" apps, i.e., low-quality, low-credibility apps (including "malware" apps) that are uploaded to app stores in the hope of quickly profiting from unsuspecting users (sometimes referred to as "spam" apps; see <https://www.theverge.com/2024/7/19/24201756/google-play-store-update-purge-low-quality-android-apps>.)

However, if we confine ourselves to the field of mental health apps, we can see that many such apps have been downloaded by substantial numbers of users. Years ago, when we began conducting background research on the marketplace prior to developing our app, we searched the Apple App Store and the Google Play Store (using terms such as "wellbeing" and "wellness") and created a spreadsheet of the most relevant apps in this area. Although we did not conduct an exhaustive search, our spreadsheet listed the apps that a user would typically encounter upon searching the stores for mental wellbeing products (including relatively obscure apps). Our spreadsheet listed 59 apps, and on 7 February 2025, we revisited this list while preparing our responses to the reviewers' comments on our protocol paper. We checked how many times each app had been downloaded from the Google store (again, download numbers for the Apple store are not publicly displayed), and of the 59 apps on the list, we found the following download numbers:

- 17 apps had over 1 million downloads;
- 12 apps had between 100,000 and 1 million downloads;
- 15 apps had between 10,000 and 100,000 downloads;
- 2 apps had between 1,000 and 10,000 downloads;

- 4 apps had between 100 and 1,000 downloads;
- 9 apps were defunct or no longer available on the Google store.
(Please see Table 1 in the Appendix of the present document.)

Another way of gauging download numbers is to consult the website that informed Lagan et al.'s (2021) extensive review (comprising 278 apps) of the mental health app marketplace, which we have cited in our protocol paper. In this review, Lagan et al. utilised the publicly available MIND database (Mobile-health Index & Navigation Database) from the website <https://apps.digitalpsych.org>, which has since been relocated to <https://mindapps.org>. On 7 February 2025, we searched this database for relevant terms (“wellbeing”, “well-being”, “well being”, “wellness”, “happy”, “happiness”, “satisfaction”, and “positive”) and created a table listing the apps retrieved from these searches (please see Table 2 in the Appendix of the present document). When we searched for “wellbeing” (and the related terms “well-being”, “well being”, and “wellness”), we found 9 apps on the MIND database. Of these apps, three were not available on the Google store (of which one app was not available on the Apple store either; the website for this app claims that it has been used by at least 250,000 people). Of the remaining six apps, all but one had been downloaded over 100,000 times. The app with fewer than 100,000 downloads was specifically designed for mothers, and it still had over 1,000 downloads. When we searched for “happy” and “happiness”, we found 7 apps on the database. Of these apps, only one was not available on the Google store. The other six all had over 100,000 downloads. Our search for the words “satisfaction” and “positive” yielded no results from the MIND database (this underscores the fact that mental health interventions, including mental health apps, often prioritise treating mental symptoms rather than promoting mental wellbeing or positive psychology, which reinforces our point that there are gaps in the market and in the literature with regard to app-based mental wellbeing interventions, especially multi-component positive psychology interventions – see the Introduction and Discussion sections of our manuscript).

In the research literature, we found a paper by Baumel et al. (2019) that analysed usage of mental health apps in a sample of market research participants from the private company SimilarWeb. The analysis suggested that such apps typically have low levels of usage, with average retention levels dropping to around 5% after 30 days (with approximately 70% of users opening the average app in the first day after downloading it). However, this paper pertained only to free apps, rather than paid apps or apps with a limited free trial period. It is easy for smartphone users to download numerous free apps, try them for a few days, and then keep browsing or choose only one of the apps to continue using in the longer term. Furthermore, free apps might tend to be relatively low in quality or lacking in activities, discouraging user retention. The characteristics of SimilarWeb's participants are also unknown. As market research participants, they may be more inclined to download higher numbers of apps and try them for shorter periods. Even so, most of the apps in Baumel et al.'s analysis had over 10,000 downloads. Even if only 5% of those downloads translated to committed users over 30 days or longer, the resulting sample for an app with 10,000 downloads would still contain 500 participants.

Overall, it seems that mental wellbeing apps are often downloaded thousands of times, and although downloading an app does not imply that the user will use the app extensively, it seems that a minority of apps are used only once after being downloaded. However, we accept the Reviewer's point that it is crucial to encourage engagement with our app, and that our study depends on recruiting sufficient numbers of participants who will use the app as instructed over the study period. In order to promote engagement and retention of

participants, we have designed the app to be as appealing and engaging as possible (e.g., with attractive graphic design, a convenient user interface, etc.), in collaboration with the app development team (Miroma Project Factory), who are experienced app developers, as well as our UAT (User Acceptance Testing) volunteers and other volunteers from the local community (see the “Public involvement” section of our manuscript). Our app has also been programmed to send reminder notifications to the user if they have not opened the app for more than 72 hours (see page 13 of the revised manuscript).

Furthermore, we expect that substantial uptake will be driven by our efforts to raise awareness of the app once it has been officially launched. As researchers who invented the app while employed at Neuroscience Research Australia (NeuRA), we have the institutional backing of NeuRA, the University of New South Wales, and the Prince of Wales Hospital, along with their media teams. When our app is officially launched, we will run a campaign to inform the public about the app and the science behind it, and to promote it to those who are seeking to increase their mental wellbeing via a digital intervention.

Beyond these measures, we are in the same position as any other app researcher who seeks ecological validity (i.e., who recruits participants “naturally” via self-initiated downloads from the app stores): We must accept the vicissitudes of the marketplace and continue recruiting participants until our target sample size has been achieved (see page 15 of the revised manuscript). Finally, when publishing the results of this study, we will of course provide statistics on the number of participants relative to the number of downloads, the levels of attrition, and all other relevant findings regarding user engagement and retention.

2) Given the first point re: app non-usage, I think a stronger argument needs to be made about why creating an app in the first place is a valuable endeavor. Is there prior work that suggests that uptake of something like this could occur at scale and be valuable. This could be further developed in the introduction.

As discussed in our previous response, there appears to be strong demand for mental health apps, with the majority of the wellbeing apps listed in the MIND database having been downloaded at least 100,000 times. It might be argued that the current offerings in the marketplace are sufficient to meet this demand, and therefore that a new app might struggle to attract a large number of users. However, we do not believe this to be the case. It is true that the market is relatively saturated with meditation and mindfulness apps, but it offers only a handful of multi-component wellbeing apps that target multiple positive psychology techniques and multiple domains of mental wellbeing. Of this small number of apps, most are not explicitly based on the peer-reviewed literature, and most have not been empirically validated. Thus, prospective users who seek wellbeing techniques beyond mindfulness or meditation have relatively few options at present. We expect that our app will stand out as a unique offering (see page 9 of the revised manuscript), given that it is a multi-component intervention, incorporates a validated measure of wellbeing (the COMPAS-W scale), is based on peer-reviewed research (see Table 1 of our manuscript), and will be empirically evaluated from the outset (unlike many other wellbeing apps, which may remain on the app stores for years without ever being evaluated, despite claims in their product descriptions that they have proven efficacy). This gives our app a level of credibility that is rare in the app marketplace.

Regarding the question of whether there is prior work to suggest that uptake of an app such as ours could occur at scale and be valuable, we have noted in the Introduction of our manuscript that there is a “research-to-retail gap”, with most mental health apps being

empirically untested. We believe this gap actually justifies our own study, because if researchers do not close the gap, then we will remain scientifically uninformed of the effects of mental health apps. Given that many such apps have been downloaded by thousands or even millions of users, it would be a fruitful opportunity for the research community to investigate their effects more extensively. Furthermore, despite the research-to-retail gap, there does exist a body of research on whether mental health apps can be valuable to users. As mentioned in the Introduction of our manuscript, there have been a number of RCTs of mental health apps which have shown benefits for the participants. For example, Howells et al. (2016) evaluated the meditation app Headspace (which has been downloaded over 10 million times), finding increases in positive affect and decreases in depressive symptoms; Bakker et al. (2018) evaluated MoodMission (over 5,000 downloads), finding increases in wellbeing and decreases in depression; Gnanapragasam et al. (2023) evaluated Foundations (now called Koa Care 360, with over 10,000 downloads), finding reductions in psychiatric symptoms; and May and Maurin (2021) reviewed Calm (over 50 million downloads), reporting a range of physical and mental health benefits.

However, we agree with the Reviewer that the feasibility of app-based interventions (in terms of both uptake and effectiveness) was not sufficiently salient in the Introduction of our manuscript, and so we have revised it to add additional details and citations regarding these issues (see pages 6-7 of the revised manuscript), including remarks about the wellbeing apps listed in the MIND database and the large numbers of downloads that most of these apps have received.

3) The authors could clarify who the target population is, and if there is a need for 'mental wellness' among this group. While there is certainly a great need for mental health services, this app is not aimed at doing this. Without this, the paper isn't providing a convincing argument that this is a worthwhile endeavor.

The target population of our app is the general population of adult users on the app store. Although we have mentioned this population throughout the Introduction section of our manuscript, we agree with the Reviewer that we did not explicitly state that this population is the target of our own app. We have revised our manuscript to make this clear (see page 10 of the revised manuscript).

Regarding the mental health needs of the target population, our app is designed to increase mental wellbeing rather than treating mental illness/symptoms (we have included measures of mental symptoms in our outcome measures in order to obtain a broader picture of the participants' mental health levels, but our primary focus is mental wellbeing as measured by the COMPAS-W scale). According to the dual-continua model of mental health (see Mason Stephens et al., 2023, cited in the Introduction of our manuscript), mental health comprises not only minimal mental illness but also optimal mental wellbeing, with the two domains sharing a relatively small amount of variance. We agree with the Reviewer that there is a great need for mental health services to treat mental illness and alleviate symptoms. However, there is also a need to address the distinct domain of mental wellbeing. Mental wellbeing is important as an end in itself, as it is defined by higher happiness, satisfaction, fulfilment, and self-compassion. Furthermore, mental wellbeing is important as a driver of other positive outcomes, such as productivity, physical health, and cognitive functioning. However, we agree with the Reviewer that our original manuscript did not point out that there is a need for increased mental wellbeing in the target population. In our revised manuscript, we have included an additional passage (with citations of key meta-analyses on wellbeing

interventions in the general population) to make and support this crucial point (see page 10 of the revised manuscript). This passage points out that the vast majority of adults in the general population fall into the “languishing” or “moderate” categories with regard to mental wellbeing, with few falling into the “flourishing” category, hence the need for mental wellbeing interventions.

4) For those who download and decide to use the app, what forms of engagement strategies might there be to ensure continuous use e.g. push notifications?

As mentioned in the Methods section of our protocol paper, our app is designed to send reminder notifications to the user if they have not opened the app for over 72 hours (see page 13 of the revised manuscript). Our app also provides the user with the option to link the app with their iPhone calendar, in order to add relevant events (e.g., based on the “positive event scheduling” activity) and to be reminded of them at the appropriate times. We neglected to mention the calendar-linkage function in our original manuscript, so we have added this information in the revised version (see page 13 of the revised manuscript). We thank the Reviewer for prompting us to notice and address this oversight.

Other than the aforementioned features, our intervention (like any other) will depend on participant compliance based on the participants’ own interest and motivation to use the app for their own perceived benefit.

5) The authors could also describe what do they do if people answer with low scores on depression, anxiety and stress scales, and how they manage risk here.

As mentioned in the Methods section of our protocol paper (see pages 16-17 of the revised manuscript), prospective participants who report moderate-to-severe psychological distress at the screening stage will not be included in our study. Users who pass the screening questions may experience higher levels of distress at a later stage, and they have the option of navigating via the app dashboard to access contact information for 24-hour mental health support services (see page 13 of the revised manuscript). These services are also mentioned in the Participant Information Statement (where participants are advised to access the services if they experience distress at any point during the study), but we did not state this fact in the original manuscript. We have now added this information to the revised version (see page 13 of the revised manuscript), and we thank the Reviewer for prompting us to detect this oversight.

Furthermore, at each measurement occasion, the survey concludes with an adverse event assessment (see the Supplementary Materials), whereby the participants can report any aspect of the survey or app that may have caused distress. This section also contains the following statement: “In the App Settings menu, we have provided contact details for Mental Health Support Services for your country, should you ever feel distressed and require additional mental health support.” This statement was accidentally omitted from the Supplementary Materials in our original submission, and we have included it in our revised submission (see page 8 of the revised Supplementary Materials; see also page 20 of the revised manuscript). We thank the Reviewer for prompting us to address this omission.

Very interesting evaluation approaches for apps are proposed. I hope the authors find this useful.

We are sincerely grateful for Dr Strudwick's insightful feedback and for the improvements we have been able to make to our manuscript as a result.

Reviewer 2:

Dr. Myoungsuk Kim, Kangwon National University

Comments to the Author:

It was a great pleasure to review your manuscript, which presents valuable insights and contributes significantly to the field. I have provided detailed comments and suggestions throughout the manuscript, which I hope will help further enhance the quality, clarity, and impact of your research.

1. Abstract: It would be beneficial to include a description of the ReNeuWell App in the abstract.

We agree with the Reviewer that the abstract should include a description of our app, and we have revised our manuscript accordingly (see page 2 of the revised manuscript).

2. Introduction: The review of prior research on mental wellbeing apps is well-conducted. However, while the limitations of existing apps are described in relation to the development of ReNeuWell, is there any explicit mention of how the newly developed app addresses these limitations? Although the app is differentiated from existing ones, it doesn't seem to clearly overcome the shortcomings of previous apps. Please elaborate on the unique features of this app and how it differentiates itself from other apps.

We agree with the Reviewer that the distinct advantages of our app were not sufficiently clear in our Introduction. We have added a passage to the Introduction (see page 9 of the revised manuscript) wherein we explicitly state how our app and our proposed RCT will address key limitations of previous app-based interventions: Our app is multi-component, offering activities across 18 areas (see Table 1 of our protocol paper), whereas most other apps offer only one or a few components; our app incorporates a validated measure of mental wellbeing (the COMPAS-W scale), whereas most other apps either do not incorporate any wellbeing measures or incorporate bespoke measures that have never been evaluated in the literature; the activities in our app are explicitly based on the peer-reviewed literature (see Table 1 of our protocol paper); and our proposed RCT will recruit participants "naturally" via self-initiated downloads from the app store, giving our study a level of ecological validity that is missing from most other studies of app-based wellbeing interventions.

3. Methods

1. The anticipated attrition rate is over 50%, which could impact the reliability of the study's results. It would be important to explore strategies to reduce the attrition rate. Please include potential measures to address this issue.

As mentioned in the Methods section of our protocol paper, our app is designed to send reminder notifications to users who have not accessed the app in over 72 hours (see page 13 of our revised manuscript). We have also designed the app to be as appealing and engaging as

possible (e.g., via attractive graphic design, a convenient user interface, etc.), in collaboration with the experienced app development team (Miroma Project Factory) and the volunteers from our UAT (User Acceptance Testing) program, as well as other volunteer testers from the local community (see the “Public involvement” section of our manuscript). Beyond these measures, there is no way to reduce attrition without compromising the ecological validity of our study: Every aspect of the study is delivered automatically via the app, to reflect the real-world context in which apps are typically used (i.e., as standalone programs that do not require the user to interact with anyone else). Our intervention, like any other, will depend on the participants’ interest and motivation to continue with the study for their own perceived benefit.

Of course, we agree with the Reviewer that attrition is a significant issue, which is why we have not only taken the aforementioned measures to promote user engagement, but also committed to compensating for possible attrition via our proposed statistical analyses, whereby we will use an intention-to-treat approach and multiple imputation (if needed) in order to account for missing data and to minimise bias in our results. Our analyses will also allow us to use baseline and midpoint data to examine possible differences between those who drop out after these time-points and those who remain in the study until the final measurement occasion, so that we can interpret our findings accurately. We have revised our manuscript to make these points explicit (see pages 21-22 of the revised manuscript), and we thank the Reviewer for bringing these matters to our attention.

2. The study will last for 12 weeks, but it might be necessary to provide justification for this duration. 12 weeks may not be sufficient to fully assess improvements in mental health.

We agree with the Reviewer that it is necessary to justify the duration of our study, and we have revised our manuscript accordingly (see page 14 of the revised manuscript). Specifically, we have noted that previous RCTs of wellbeing apps (that showed efficacy in boosting mental wellbeing) had similar or even shorter durations. For example, the study by Howells et al. (2016) lasted 10 days, the study by Bakker et al. (2018) lasted 30 days, the study by Gnanapragasam et al. (2023) lasted 8 weeks, the study by Mak et al. (2018) lasted 16 weeks, the study by Economides et al. (2018) lasted one month, the study by Golec de Zavala (2024) lasted 6 weeks, and the study by van Agteren et al. (2021) lasted 7 days.

Another justification for the duration of our study is that users may not persist in using a mental health app if they do not subjectively experience benefits in a relatively short period of time. After a few weeks of usage, if a user would not report any benefits, they might stop using the app and perhaps download an alternative mental health app instead. We have revised our manuscript to mention this (see page 14 of the revised manuscript).

However, if our study demonstrates app efficacy within the 12-week study period, we will be open to testing our app over longer durations in subsequent research.

3. Since only participants without mental health issues are included in this study, there may be limitations in evaluating the actual effectiveness of the app. Specifically, because participants do not have mental health problems, the effects of the app may be underestimated or deemed ineffective. It may be beneficial to include participants with mental health conditions to better assess the app's effectiveness.

In our revised manuscript, we have added a passage to the Introduction (see page 10 of the revised manuscript) to justify the use of mental wellbeing and positive psychology interventions in non-clinical populations (such as the general population of adult app users from which we will recruit the sample for our study). This passage includes citations of key meta-analyses that have shown the effectiveness of such interventions in both non-clinical and clinical populations. Although the benefits of these interventions are often greater in clinical populations, we wish to demonstrate efficacy in users without mental health problems in order to maximise the ecological validity of our study (given that most members of the general population are not currently experiencing significant mental illness, and we hope to demonstrate that our app can benefit the typical user who is mostly free of mental illness symptoms). We are open to testing our app in clinical populations in future studies, but for the present study, we have received ethical approval to test it only in participants without substantial baseline levels of mental distress, given that our app is a newly developed intervention.

References

- Bakker D, Kazantzis N, Rickwood D, Rickard N. A randomized controlled trial of three smartphone apps for enhancing public mental health. *Behav Res Ther.* 2018;109:75-83.
- Baumel A, Muench F, Edan S, Kane JM. Objective user engagement with mental health apps: systematic search and panel-based usage analysis. *J Med Internet Res.* 2019;21(9).
- Economides M, Martman J, Bell MJ, Sanderson B. Improvements in stress, affect, and irritability following brief use of a mindfulness-based smartphone app: a randomized controlled trial. *Mindfulness.* 2018;9:1584-1593.
- Gnanapragasam SN, Tinch-Taylor R, Scott HR, Hegarty S, Souliou E, Bhundia R, et al. Multicentre, England-wide randomised controlled trial of the 'Foundations' smartphone application in improving mental health and well-being in a healthcare worker population. *Br J Psychiatry.* 2023;222(2):58-66.
- Golec de Zavala A, Keenan O, Ziegler M, Ciesielski P, Wahl JE, Mazurkiewicz M. App-based mindfulness training supported eudaimonic wellbeing during the COVID19 pandemic. *Appl Psychol Health Well Being.* 2024;16(1):42-59.
- Howells A, Ivtzan I, Eiroa-Orosa FJ. Putting the 'app' in happiness: A randomised controlled trial of a smartphone-based mindfulness intervention to enhance wellbeing. *Journal of Happiness Studies: An Interdisciplinary Forum on Subjective Well-Being.* 2016;17:163-85.
- Lagan S, D'Mello R, Vaidyam A, Bilden R, Torous J. Assessing mental health apps marketplaces with objective metrics from 29,190 data points from 278 apps. *Acta Psychiatr Scand.* 2021;144(2):201-10.
- Mak WW, Tong AC, Yip SY, Lui WW, Chio FH, Chan AT, et al. Efficacy and Moderation of Mobile App-Based Programs for Mindfulness-Based Training, Self-Compassion Training, and Cognitive Behavioral Psychoeducation on Mental Health: Randomized Controlled Noninferiority Trial. *JMIR Ment Health.* 2018;5(4):e60.

Mason Stephens J, Iasiello M, Ali K, van Agteren J, Fassnacht DB. The Importance of Measuring Mental Wellbeing in the Context of Psychological Distress: Using a Theoretical Framework to Test the Dual-Continua Model of Mental Health. *Behav Sci.* 2023;13(5):436.

May AD, Maurin E. Calm: a review of the mindful meditation app for use in clinical practice. *Families, Systems, & Health.* 2021;39(2):398-400.

van Agteren J, Bartholomaeus J, Steains E, Lo L, Gerace A. Using a technology-based meaning and purpose intervention to improve well-being: A randomised controlled study. *Journal of Happiness Studies: An Interdisciplinary Forum on Subjective Well-Being.* 2021;22:3571-91.

Appendix

Table 1. Numbers of downloads across 59 apps from our market research.

App name	Downloads from the Google Play store (as of 7 February 2025)
1 Giant Mind: Learn Meditation	10K+
7 Cups	1M+
AbleTo (formerly Joyable)	50K+
Anxiety Reliever: Mental Health Support	defunct
Bloom: Better You / Bloom: Meditation & Sleep	10K+
Breathe2Relax	10K+
Breethe – Meditation & Sleep App	1M+
Breeze: Start Self-Discovery	1M+
Calm – Meditate, Sleep, Relax	50M+
CBT Companion: Therapy Journal	Apple App Store only
Check-in – Beyond Blue	defunct
Clarity – CBT Thought Diary	500K+
Daylio	10M+
Dr.Mind – Mental Health Tests	10K+
Exhale: Guided Breathwork	1K+
Fabulous Daily Routine Planner	10M+
FreeCBT	10K+
Guided Wellbeing Library	1K+
Happify	500K+
HeadGear (by the Black Dog Institute)	defunct
Headspace: Meditation & Sleep	10M+
Healthy Minds Program	100K+
HelloMind	50K+

Insight Timer – Meditation App	5M+
MindDoc: Mental Health Support (formerly MoodPath)	1M+
MindShift CBT – Anxiety Relief	500K+
MoodKit	defunct
MoodMission – Cope with Stress	5K+
MoodNotes	5K+
MoodPrism	100+
MoodSpace	100+
MoodTools – Depression Aid	100K+
My Life Meditation	500+
My Mood Tracker: Mood Control	10K+
myStrength by Teladoc Health	100K+
Only Human	500+
Pause – Guided Meditation & relaxing sleep stories	100K+
Peerhose: Mental Health Support Community	defunct
ReachOut WorryTime	defunct
Relax Change Create Meditation	10K+
Relax with Andrew Johnson	10K+
Sanvello: Anxiety & Depression (formerly Pacifica)	defunct
Shine: Calm Anxiety & Stress / Shine: Self-care & Meditation	defunct
Simple Habit: Meditation	1M+
Sleep Sounds – Relax Music	5M+
Smiling Mind: Mental Wellbeing	1M+
Stresscoach: Reduce Anxiety (formerly Pocketcoach)	50K+
SuperBetter: Mental Health	100K+
The Mindfulness App	1M+
Thinkladder – Self-awareness	50K+
ThinkUp – Daily Affirmations	100K+
Thrive	10K+
Tide – Sleep & Meditation	1M+
Unwinding Anxiety	100K+
Waking Up: Beyond Meditation	1M+
What's Up? – Mental Health App	500K+
Woebot: Your Self-Care Expert	500K+
Wysa: Anxiety, therapy chatbot	1M+
Youper – CBT Therapy Chatbot	1M+

Table 2. Numbers of downloads across 16 wellbeing/happiness apps from the MIND database.

App name	Downloads from the Google Play store (as of 7 February 2025)
Search terms: wellbeing, well-being, well being, wellness	
Betwixt – The Mental Health Game	100K+
Intuitive Eating Diary – Munch	Apple App Store only
Kai – Your wellness companion	Not available ^a
Nourish (wellbeing for mums)	1K+
Remente: Self Care, Wellbeing	1M+
Sharecare: Health & Well-being	500K+
Smiling Mind: Mental Wellbeing	1M+
ThinkUp – Daily Affirmations	100K+
Thunderbird Wellness: Supporting Indigenous Wellness	Apple App Store only
Search terms: happy, happiness	
365 Gratitude Journal	500K+
ACT Companion: Happiness Trap	Apple App Store only
Action for Happiness: Get Tips	100K+
Happify	500K+
Happyfeed: Gratitude Journal	50K+
HeadApp! Headache Diary	100K+
Therapyside – Online Therapy	100K+

^a Not available on Google nor Apple stores (website claims 250K users).

VERSION 2 - REVIEW

Reviewer **1**

Name **Strudwick, Gillian**

Affiliation **Centre for Addiction and Mental Health, Information Management Group**

Date **07-Mar-2025**

COI

The responses back to my comments demonstrate that the authors dont know this field well despite their lengthy responses. I think its fine to publish this protocol but I am not sure how useful it will be for the field.

Reviewer	2
Name	Kim, Myoungsuk
Affiliation	Kangwon National University, College of Nursing
Date	12-Mar-2025
COI	

It was a great pleasure to review your manuscript, which presents valuable insights and contributes significantly to the field. I have provided detailed comments and suggestions throughout the manuscript, which I hope will help further enhance the quality, clarity, and impact of your research.

While the revisions have been made, I would like to request further revisions on the following part.

1) It is necessary to provide a clear rationale for why mental health apps should be developed for the general population. Existing apps have often been developed with the aim of preventing and promoting mental health for at-risk individuals who already experience mental health issues. However, there is insufficient justification for developing such an app for the general population, particularly those without existing mental health concerns. This app is designed to evaluate its effectiveness in individuals without mental health issues, which could potentially result in the app showing no effect. It is important to acknowledge this possibility and clearly explain why targeting a population without mental health issues is still valuable.

2) A more in-depth discussion of the strengths and weaknesses of existing apps is needed, and there is a lack of evidence regarding how the app being developed compares in terms of its advantages and effectiveness over existing apps. Therefore, it would be beneficial to provide a more detailed explanation of this aspect.